# MemoryLLM: Plug-n-Play Interpretable Feed-Forward Memory for Transformers

**Ajay Jaiswal** [1]   **Lauren Hannah** [1]   **Han-Byul Kim** [1]   **Duc Hoang** [1]   **Arnav Kundu** [1]   **Mehrdad Farajtabar** [1]
**Minsik Cho** [1]

## Abstract

Understanding how transformer components operate in LLMs is important, as it is at the core of recent technological advances in artificial intelligence. In this work, we revisit the challenges associated with interpretability of feed-forward modules (FFNs) and propose **MemoryLLM**, which aims to decouple FFNs from self-attention and enables us to study the decoupled FFNs as context-free token-wise neural retrieval memory. In detail, we investigate how input tokens access memory locations within FFN parameters and the importance of FFN memory across different downstream tasks. MemoryLLM achieves context-free FFNs by training them in isolation from self-attention directly using the token embeddings. This approach allows FFNs to be pre-computed as token-wise lookups (ToLs), enabling on-demand transfer between VRAM and storage, additionally enhancing inference efficiency. We also introduce Flex-MemoryLLM, positioning it between a conventional transformer design and MemoryLLM. This architecture bridges the performance gap caused by training FFNs with context-free token-wise embeddings.

## 1. Introduction

Large language models (LLMs) are *omnipresent*, demonstrating rapidly evolving capabilities across critical domains such as healthcare, finance, education, and mobility. Modern LLMs (Grattafiori et al., 2024; Liu et al., 2025; 2024; Jiang et al., 2024; Adler et al., 2024; Gunter et al., 2024) are sequential stacks of transformer blocks in association with embedding layers that project textual input to a latent representation for processing, followed by projection back to tokens. Each transformer block is constituted by two key computationally expensive components: self-attention (Attn) and feed-forward (FFN) modules. Self-attention modules within LLMs are considered the key component in transformer blocks, constructing representations of the current input by aggregating relevant information from the context (Sukhbaatar et al., 2019). Numerous works have studied the role and optimization (Vig, 2019; Xiao et al., 2023; 2024; Clark et al., 2019) of self-attention by investigating token-level attention patterns generated during input text processing.

In contrast, FFNs, while holding approximately two-thirds of the LLMs' parameters, have been relatively underexplored, and their roles in information processing flow deserve in-depth studies. One potential reason for limited investigation can be attributed to the existence of a tightly intertwined relationship between FFNs and self-attention in modern LLMs: FFNs consume a non-interpretable additive mixture of self-attention output and residual stream (Figure 1a) that makes the attempt to study the contributions of FFNs non-trivial. Some notable prior works (Geva et al., 2021; 2022; Dar et al., 2023) have attempted to establish, within the limited scale of GPT-2, that FFNs in pretrained LLMs serve as neural key-value memory over textual input patterns such as n-grams or semantic topics. However, such interpretations suffer from two critical shortcomings: (1) dependency on multiple forward and backward passes with calibration dataset fed through the model followed by a careful mining of relevant input phrases with manual annotation; (2) inability to discretely define interpretable queries for key-value memory access within FFNs.

Motivated by the interpretability challenges associated with FFNs in LLMs, we pose a critical question: *How can we disentangle FFNs from self-attention to encode deterministic memory mapped to a finite human-interpretable vocabulary?* Unlike previous attempts which focus on investigating *de facto* FFNs of pretrained LLMs, we adopt an alternative approach: designing and training from scratch a novel transformer architecture. Our main goal is to enforce

*Equal contribution  [1]Apple. Correspondence to: Ajay Jaiswal <ajaiswal23@apple.com>.

clear separation between self-attention and FFNs, in order to clarify the FFN's role as context-free retrieval memory over an interpretable, finite set of tokens.

In this work, we propose **MemoryLLM**, which makes a dramatic simplification of a conventional transformer architecture that entails training self-attention and FFNs modules in independent of and parallel to each other. **MemoryLLM** features a interpretability-focused architecture (Figure 1b) where self-attention heads are trained in conventional fashion using the incoming residual stream, while FFNs are trained in isolation directly on *context-free and token-indexed embedding vectors*. This isolation allows us to build discrete token-level retrieval memory in FFNs. MemoryLLM enables two main advantages:

- **FFN Interpretability:** With a fixed and discrete token-wise query space, FFNs' interpretation as neural key-value retrieval memories can be well-studied for each token in a model's vocabulary.
- **LLM Efficiency:** With static token embedding-based training directly from embedding layer, FFN modules in MemoryLLM can be pre-computed and offloaded to storage devices.

We train-from-scratch MemoryLLM at different parameters scale (250M, 750M, and 1B) to study its capabilities. Our contributions are as follows:

- We revisit the challenge in investigating the role of FFNs through a novel lens of token-indexed neural retrieval memory. We present a *TKV (token-key-value) framework* to investigate how FFNs construct a persistent context-free memory over the model's vocabulary.
- Building on top of prior tools (Geva et al., 2021; 2022), we explore the *spatial perspective* of token-indexed memory where lexically and semantically similar query tokens tend to access similar memory location within FFNs for retrieval.
- We find that FFNs in MemoryLLM acts as reservoirs of token-level parametric knowledge learned directly from training data. They play a **dominant** role in retrieval-based tasks in comparison to inferential or logical thinking tasks.
- MemoryLLM also addresses the memory and computational bottleneck of LLMs at inference by facilitating pre-computation of FFNs as **token-wise lookups (ToLs)** and **plug-n-play (PnP)** memory transfer from storage devices under resource constraints.
- To closely match performance of conventional LLMs, we introduce **Flex-MemoryLLM**. This architecture is positioned between standard transformer blocks and MemoryLLM, effectively bridging the gap by splitting FFNs parameters among context-aware and context-free FFN modules.

## 2. MemoryLLM: LLMs with Interpretable Token-Indexed Feed-Forward Memory

### 2.1. Motivation

Modern LLMs are composed of a sequential stacking of self-attention and feed-forward modules. Figure 1(a) illustrates a residual information flow perspective (Elhage et al., 2021) in a conventional LLM, where at transformer layer $L$, the self-attention module takes the snapshot $X_L$ of residual stream and transforms it with contextual information before adding it back to the stream. Next, the FFN module reads the residual stream, which now carries the contextual information, processes it and adds it back to residual stream, resulting in $X_{L+1}$ for next layer. In LLMs, self-attention is often regarded as the key driver of success, attracting significant research into its functionality and optimization. In contrast, FFNs, which constitute approximately two-thirds of total parameters, have been *underexplored*, particularly regarding how they process the residual stream and store information during training. In this two-step mechanism within a conventional transformer block, FFNs consume a *non-interpretable latent input*, which is an additive mixture of self-attention output and residual stream. This representation dynamically evolves between layers, and is the primary bottleneck in understanding FFN function during token processing.

Earlier works (Geva et al., 2021; 2022) argue that feed-forward layers emulate neural key-value memories. They support this claim by mapping keys within FFNs to manually annotated textual patterns in the training data. However, mapping keys of an intermediate transformer layer $L$ within an FFN back to initial input tokens is neither straight-forward nor optimal. The residual stream changes significantly between $X_0 \rightarrow X_L$ with a complex amalgamation of contextual information from previous layers. Consequently, the query space to investigate FFNs' emulated key-value memory remains a non-interpretable latent input, with context-dependent properties. To address this, we decouple FFNs from the residual stream, enabling a deterministic investigation of FFNs as reservoir of key-value memory accessible with a finite query space.

In the next subsection, we present **MemoryLLM**. This architecture allows us to study FFNs in isolation of self-attention, where each token id from an input prompt can be mapped to *static* indices within context-free token-level memory.

### 2.2. Architecture Design

#### 2.2.1. PRELIMINARIES: CONVENTIONAL LLMS

An input text $T = \{t_1, t_2, ..., t_M\}$ of $M$ tokens is transformed to embedding vectors $X_0^{M \times d}$ with an embedding matrix $E \in R^{|V| \times d}$ over a vocabulary space $V$. A self-

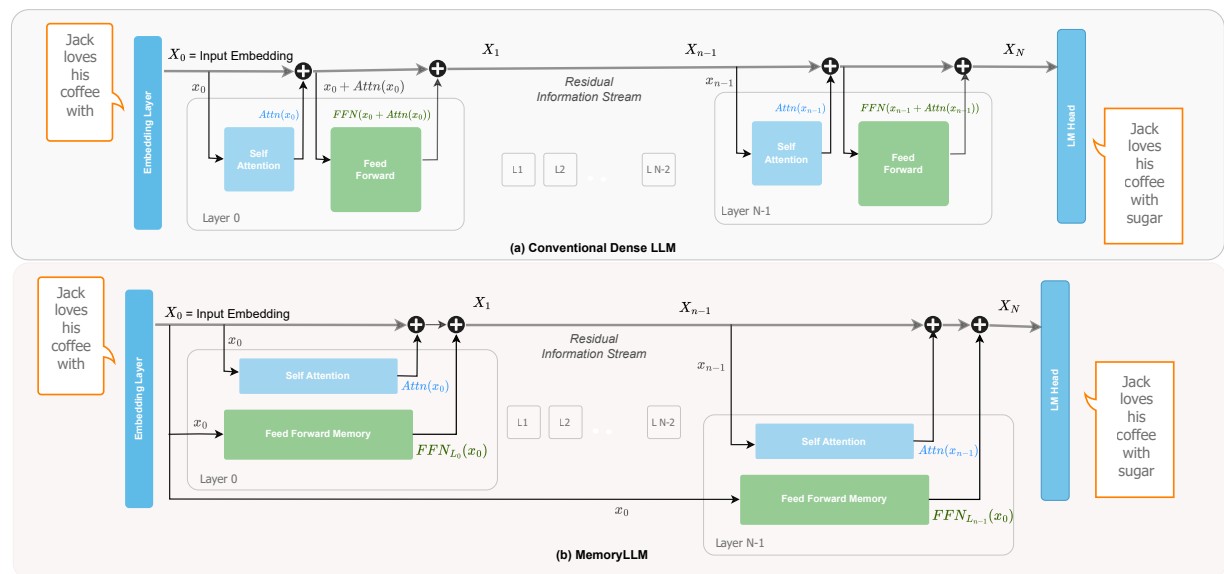

*Figure 1.* **Architecture comparison of Conventional Transformer** (base) **v/s MemoryLLM with Residual Stream Perspective:** (a) FFN input in conventional transformers is a sequential and non-interpretable latent snapshot of a residual stream, including prior self-attention module output; (b) MemoryLLM decouples FFNs across all transformer blocks completely from self-attention modules and trains them in isolation of the residual stream, directly on token-indexed input embeddings.

attention module at transformer layer $L$ is parameterized with $W_Q, W_K$, and $W_V$ weight matrices. The contextualized token representation is computed by:

$$\text{Attn}(X_L) = \text{softmax}\left(\frac{X_L W_Q^\top (X_L W_K^\top)^\top}{\sqrt{d_k}}\right) X_L W_V^\top. \quad (1)$$

Next, SwiGLU (Shazeer, 2020) based FFNs in modern LLMs are parameterized with three matrices with $K$ as intermediate expansion dimension: $W_{Up}, W_{Down}$, and $W_{Gate}$. In a conventional transformer layer, FFN output is an additive mixture of self-attention and residual stream ($\tilde{X}_L$):

$$\tilde{X}_L = X_L + \text{Attn}(X_L), \quad (2)$$

$$\text{FFN}(\tilde{X}_L) = \left(\left(\tilde{X}_L W_{Up}^\top\right) \odot \text{SiLU}\left(\tilde{X}_L W_{Gate}^\top\right)\right) W_{Down}^T \quad (3)$$

### 2.2.2. MEMORYLLM

In this section, we describe the architecture design of MemoryLLM, which disentangles the sequential dependence of FFN modules from self-attention output and the residual stream (Equation 2,3) in a transformer layer. MoLE (Jie et al., 2025) illustrates that in mixture-of-experts (MoE), the majority of experts can be trained directly with token-level input embeddings. However, MoLE's expert computation remains conditional on contextual information from a router trained with self-attention output. Here, we explore the potential of a dramatic simplification in conventional transformer layer by parallelizing the context-aware self-attention module with context-free FFN computation, independent of the residual stream and self-attention.

More specifically, we propose to train all feed-forward modules $\{\text{FFN}_{L_0}, \text{FFN}_{L_1}, ..., \text{FFN}_{L_{N-1}}\}$ across an $N$ layer LLM, *directly* with token-indexed context-free embedding vectors $X_0 \in R^{M \times d}$ generated out of $M$ tokens IDs from tokenizer for a given input text. Figure 1(b) presents the architecture design of MemoryLLM where the self-attention module is trained in a conventional fashion (Equation 1) with incoming residual stream, while the FFN module from layer $L$ is computed as follows:

$$\hat{X}_0 = \text{LayerNorm}_L(X_0), \quad (4)$$

$$\text{FFN}(\hat{X}_0) = \left(\left(\hat{X}_0 W_{Up}^\top\right) \odot \text{SiLU}\left(\hat{X}_0 W_{Gate}^\top\right)\right) W_{Down}^T \quad (5)$$

Despite all FFNs across each transformer block receiving the same input ($\hat{X}_0$) during training, we found that having an independent layer norm ($LN_L$) for each $\text{FFN}_L$ significantly helps in convergence. Overall, for a transformer layer $L$ with incoming residual stream $X_L$, the computation for outgoing residual stream $X_{L+1}$ is:

**MemoryLLM**

$$X_{L+1} = X_L + \text{Attn}(X_L) + \text{FFN}(X_0) \quad (6)$$

Since the embedding layer output ($X_0$) is exclusively determined by the tokenizer's unique token IDs, the inputs to all the FFNs in MemoryLLM are static and drawn from a set limited to the size of vocabulary ($|V|$) of the model, regardless of phase (training or inference). This unique design not only permits a context-free token-level information storage

within FFNs but also facilitates a **plug-n-play** flexibility for a dynamic size LLM, where FFN from any transformer layer $L$ can be removed depending on VRAM constraints without disrupting the residual information flow.

## 2.3. Key Benefits: Interpretability and Efficiency

### 2.3.1. INTERPRETABILITY: TKV FRAMEWORK

In this section, we address the three key limitations of existing works (Geva et al., 2021; 2022; Sukhbaatar et al., 2019; Nichani et al., 2024): (1) the relationship between FFN memory locations and non-interpretable contextualized latents of input prefix tokens, which can significantly shift based on context; (2) the laborious reverse-engineering process required to manually mine input prefixes from calibration training data based on FFN key activations; and (3) the underexplored influence of how key-value FFN memory on downstream task performance. In modern LLMs (*e.g.,* LLaMa and GPT variants), SwiGLU based feed-forward modules are composed of three parameter matrices: up-projection ($W_{Up}$), gate-projection ($W_{Gate}$) and down-projection ($W_{Down}$) matrices.

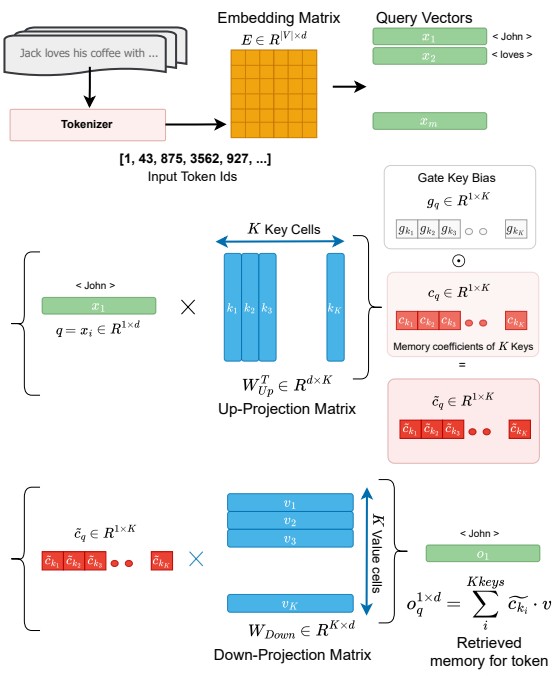

*Figure 2.* **TKV Framework:** Input text is tokenized into discrete token IDs as context-free query vectors for FFN memory cells. $W_{Up}$ and $W_{Down}$ projection matrices emulate the behavior of *Keys* and *Values* while the $W_{Gate}$ matrix can be interpreted as a reweighting function of token memory coefficients.

To overcome these interpretability limitations, we develop TKV (token-key-value) framework for FFNs that propose interpreting the *up-projection (key) and down-projection (value)* matrices as **neural retrieval memory** (Sukhbaatar

et al., 2019; Geva et al., 2021), containing $K$ key-value pairs (memory cells). In this framework, the *gate-projection* acts as learned reweighting function for keys during pretraining. Specifically, the gate-projection determines how strongly each memory cell of the FFN memory is amplified or suppressed. Figure 2 presents a detailed overview of our TKV framework. A text sequence is transformed into token-level query vectors ($x_i$) accessing FFN neural memory (represented with $W_{Up}$ and $W_{Down}$ matrices) in a context-free fashion, generating retrieved output ($o$). This is added to the residual stream (Figure 1) in parallel to self-attention. For a given query vector $q$ corresponding to token $t$, the neural memory retrieval process within FFN is a two step process:

**Step I:** Estimate the memory cell coefficient ($c_{k_i}$) corresponding to all keys $k_i$ represented as column vectors in $W_{Up} \in R^{K \times d}$ matrix with a *dot product* between $q$ and $W_{Up}^{\top}$. Intuitively, $c_{k_i}$ *can be perceived as a importance score of key $k_i$ for query vector $q$.* In SwiGLU based FFNs, $c_{k_i}$ is further elementwise reweighted with $g_{k_i}$ to amplify or suppress some specific keys:

$$\tilde{c}_{k_i} = (q^{1 \times d} \cdot W_{Up_{[:,k_i]}}^{\top}) \times g_{k_i}. \tag{7}$$

**Step II:** For a query $q$, the retrieved memory output is a $c_{k_i}$−weighted linear combination of $K$ value vectors $v_i$, which are represented as row vectors in down-projection matrix ($W_{Down}$):

$$\text{FFN}(X_0^q) = o_q^{1 \times d} = \sum_{i}^{K} \tilde{c_{k_i}} \cdot v_{k_i}. \tag{8}$$

The TKV framework of MemoryLLM addresses the challenge of undefined input query prefixes in prior works by a defining a finite set of human-interpretable query vectors from the vocabulary token IDs. This eliminates the need for reverse manual annotation of training data to study the relationship between input prefixes and corresponding memory cells within FFNs. In Section 3, we will investigate the token-level spatial properties of memory cells and their importance in downstream tasks.

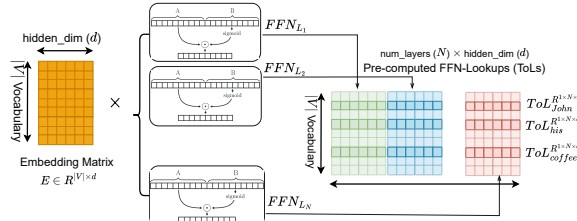

*Figure 3.* **FFNs as Pre-computed Token-wise Lookups:** Outputs corresponding to each vocabulary tokens for all FFN modules across $N$ transformer blocks can be pre-computed offline and stored as static token lookups (ToLs) in storage devices.

### 2.3.2. EFFICIENCY: FFNS AS PRE-COMPUTED LOOKUPS

In modern LLMs, FFNs account for approximately two-thirds of the model parameter budget, creating significant

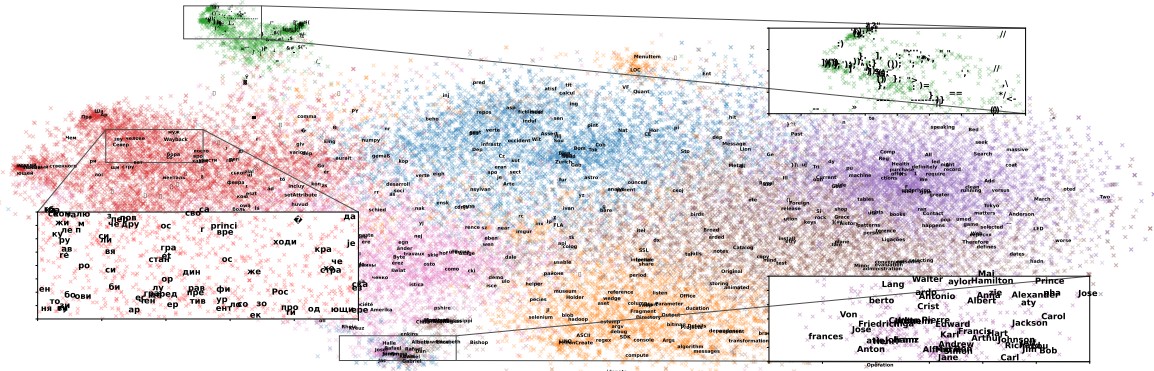

*Figure 4.* **Semantically Similar Tokens Build Memory Outputs with Similar Keys:** t-SNE plot with K-Means clustering of $c_k$ vectors, which represent each key's contribution to memory outputs, yields clusters of tokens with semantically similar properties.

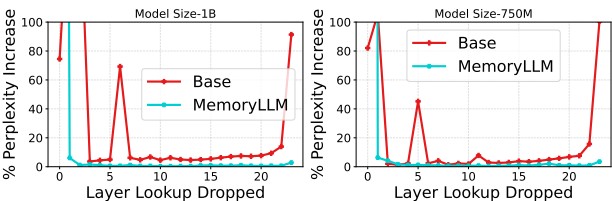

*Figure 5.* Percentage increase in perplexity when FFN computation corresponding to layer $L$ is dropped in Base and MemoryLLM.

computational and VRAM overheads that limit deployment in resource-constrained settings. In contrast to conventional designs, FFNs in MemoryLLM perform inference with static, context-free embedding outputs. This allows for a pre-computation of FFN modules over all vocabulary and off-loading into storage for asynchronous communication. Hence, MemoryLLM addresses both memory and computational constraints while improving interpretability.

Figure 3 illustrates how to pre-compute FFNs **one-time** and build a **static** lookup table for each vocabulary token ID across all transformer layers, which can be stacked horizontally and offloaded to storage devices for asynchronous prefetch during inference. Mathematically, for each token embedding $x_{t_i}^{1 \times d}$ associated with token $t_i$ for $N$-layer LLM, we generate ToLs as:

$$\text{ToL}_{x_{t_i}}^{1 \times (N \times d)} = \textbf{Concat}_{k=0}^{N-1}\left\{\text{FFN}_{L_k}(x_{t_i}), \text{ dim=1}\right\}. \quad (9)$$

**On-demand Plug-n-Play ToLs:** Our ToL formulation depends on token embeddings without any connection to intermediate features and access to contextual information. In addition to reduced computational cost per token with pre-computed ToLs, MemoryLLM presents a two fold on-demand plug-n-play design:

- ***Token-distribution follows Zipf's law:*** The token distribution of modern LLM-generated content adheres to Zipf's law irrespective of tokenizer (He et al., 2025; Zhemchuzhina et al., 2022). Therefore, VRAM re-

quirements for MemoryLLM can be significantly reduced by caching a fraction of ToLs for frequent tokens, and loading less frequently used ToLs from storage.

- ***Non-uniform importance of ToLs across layers:*** Prior works (Yin et al., 2023; Jaiswal et al., 2024a;b) have established that not all layers contributed equally to the model performance. Figure 5 shows that FFN contribution to MemoryLLM performance drops notably after first few layers, while FFNs across a conventional base LLM shows a non-uniform U-shaped behavior due to disruption in residual flow. This suggests that ToLs of later FFN layers can be offloaded permanently from VRAM with minimal disruption in residual flow.

During inference, the output of layer $L$ for $\{t_1, t_2, ..., t_M\}$ tokens in MemoryLLM can be written as:

$$X_{L+1} = X_L + \text{Attn}(X_L) + \textbf{ToL}_{[\{t_1,t_2,...,t_M\},L,:]} \quad (10)$$

Note that based on the aforementioned on-demand plug-n-play policy, ToL of a token id $t_i$ can be loaded into VRAM if there is a cache miss with a suitable caching policy.

## 3. Empirical Study of FFN Neural Memory

### 3.1. Spatial Distribution of Neural Memory in FFNs

In section 2.3.1, we developed the TKV framework that emulate FFNs as neural token-wise retrieval memory with $K$ key-value pairs. To build the output of memory ($o_q$) for a query token vector $q$, each element $c_{k_i}$ within the $c_k$ vector can be interpreted as proxy indicator of how much $k_i$ will contribute to $o_q$. In an ideal setting, $c_k$ remains a one-hot vector where a single key among $K$ keys build the output $o_q$ while in worst-case scenario, $c_k$ is a uniform vector where $o_q$ is average of the values corresponding to all $K$ keys. Here, we build on top of the prior works (Geva et al., 2021; 2022) for a large-scale LLaMa-3.1 tokenizer (Grattafiori et al., 2024), and ask: *Is there a relationship*

*Table 1.* **Controlled investigation of FFNs across tasks:** Reducing the contribution of FFNs with gradually decreasing ($\alpha$) hurts tasks that heavily rely on recall or retrieval of known information *relatively more* than reasoning or logical thinking tasks.

| Alpha | Recall/Retrieval Dominated Tasks | | | | Logical/Reasoning Dominated Tasks | | | |
|---|---|---|---|---|---|---|---|---|
| | Wikitext-2 ($\downarrow$) | LAMBDA($\uparrow$) | SiQA($\uparrow$) | ARC-Easy($\uparrow$) | HellaSwag($\uparrow$) | Winogrande($\uparrow$) | BoolQ($\uparrow$) | PIQA($\uparrow$) |
| 1.0 | 24.348 | 0.3613 | 0.7830 | 0.5156 | 0.3724 | 0.5233 | 0.6205 | 0.7018 |
| 0.9 | $24.6518_{+1.24\%}$ | $0.3643_{+0.82\%}$ | $0.7780_{-0.64\%}$ | $0.5043_{-2.19\%}$ | $0.3749_{+0.68\%}$ | $0.5193_{-0.76\%}$ | $0.6177_{-0.45\%}$ | $0.7035_{+0.24\%}$ |
| 0.8 | $25.5173_{+4.80\%}$ | $0.3546_{-1.87\%}$ | $0.7260_{-7.28\%}$ | $0.4414_{-14.40\%}$ | $0.3750_{+0.70\%}$ | $0.5178_{-1.06\%}$ | $0.6159_{-0.74\%}$ | $0.7062_{+0.63\%}$ |
| 0.7 | $27.4568_{+12.76\%}$ | $0.3004_{-16.86\%}$ | $0.6700_{-14.43\%}$ | $0.4168_{-19.16\%}$ | $0.3694_{-0.79\%}$ | $0.4988_{-4.68\%}$ | $0.6165_{-0.64\%}$ | $0.7057_{+0.55\%}$ |
| 0.6 | $32.6135_{+33.94\%}$ | $0.2002_{-44.59\%}$ | $0.6620_{-15.45\%}$ | $0.3949_{-23.40\%}$ | $0.3614_{-2.96\%}$ | $0.4901_{-6.34\%}$ | $0.6135_{-1.14\%}$ | $0.6980_{-0.54\%}$ |
| 0.5 | $53.7072_{+120.57\%}$ | $0.1086_{-69.93\%}$ | $0.5380_{-31.29\%}$ | $0.3170_{-38.52\%}$ | $0.3300_{-11.38\%}$ | $0.5012_{-4.23\%}$ | $0.6131_{-1.18\%}$ | $0.6801_{-3.09\%}$ |

*between semantically similar tokens and the specific keys they activate to generate memory outputs?*

To answer this question, we perform K-means clustering on $c_k$ vectors corresponding to all $|V|$ vocabulary tokens to identify if semantically similar tokens have similar importance score ($c_{k_i}$) for certain keys and can be clustered together. Figure 4 shows K-means clustering results of $c_k$ vectors from layer $L_0$, corresponding to all vocabulary tokens in MemoryLLM-1B. Surprisingly, we found well-formed clusters that capture multiple human-interpretable cues such as *punctuation marks, personal names, geographical locations, linguistic properties*. This finding *positively* supports the conjecture that semantically similar tokens tend to build final memory outputs with similar keys. This property provides opportunities for researchers to explore knowledge editing and injection, or toxicity suppression with targeted alteration of certain keys in interpretable fashion.

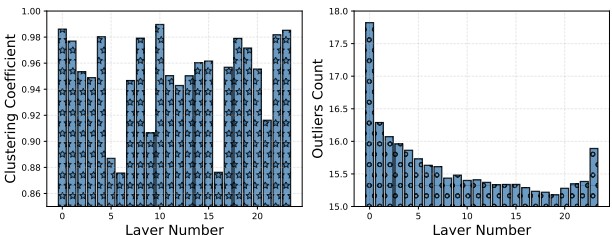

*Figure 6.* (a) Clustering coefficient for $c_k$ vectors from FFN memory across 24 layers of MemoryLLM-1B. (b) Average outliers count in $c_k$ vectors corresponding to each vocabulary token.

Next, we study how this clustering behavior translates across other layers of our MemoryLLM-1B model and empirically quantify it with the clustering coefficient (CC). In Figure 6(a), we find a high CC value across all layers in the model checkpoint, despite a slight decrease across some middle layer. Since, outlier coefficients dominate in building memory output, we study the average number of outlier coefficients cross $c_k$ vectors of 128k vocabulary tokens. Figure 6 (b) shows that terminal layers tend to have a higher number of outlier keys, which predominantly contribute towards output formation of FFNs. This hints at superior token-level information convergence within limited keys.

### 3.2. Probing FFN Memory Across Downstream Tasks

MemoryLLM design focuses on decoupling FFNs from the residual flow and allows us to study the impact of FFN

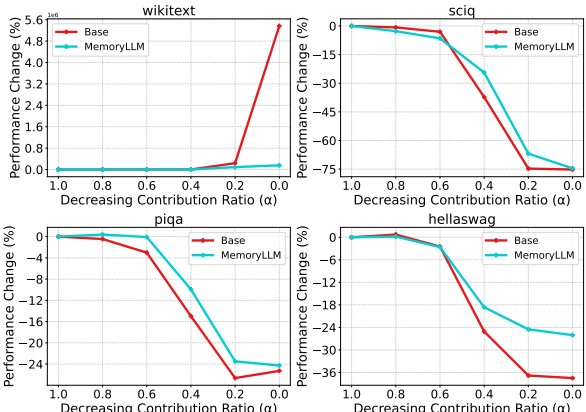

*Figure 7.* Model performance comparison with regulated contribution of FFNs in MemoryLLM and conventional base checkpoints.

*Table 2.* Perplexity comparison of conventional LLMs (base-250M & 750M) wrt. MemoryLLM with 50B training tokens.

| | base-750 | MemoryLLM | MemoryLLM | base-250 |
|---|---|---|---|---|
| **Active Params** | 737M | 402M | 245M | 265M |
| **Total Params** | 737M | 1208M | 737M | 265M |
| C4 | 19.730 | 20.933 | 22.079 | 23.190 |
| Wikitext-2 | 25.491 | 27.258 | 29.976 | 32.220 |

memory in isolation on model performance.

**First**, we start with establishing that MemoryLLM is well able to disentangle the tight coupling of FFNs within the residual flow. Following the formulation of MemoryLLM in Equation 6, we designed an experiment where we control the contribution of FFNs with an interpolation scaler as $\alpha \times \text{FFN}(X_0)$ for MemoryLLM and conventional base model checkpoints. Figure 7 illustrates how a decreasing contribution ratio of FFN memory has a relatively lesser impact on MemoryLLM performance degradation in comparison to conventional LLM designs. This behavior can be explained with the unique design choice of MemoryLLM, which **doesn't disrupt** the residual flow as significantly as base model and provides a favorable setting for post-training model compression techniques.

**Second**, we ask: *Does the influence of FFNs, viewed as token-indexed key-value memories, remain consistent across different tasks?* To investigate, we consider two broad categories of tasks: (a) tasks which heavily rely on *recall or retrieval* of known information — things that are explicitly

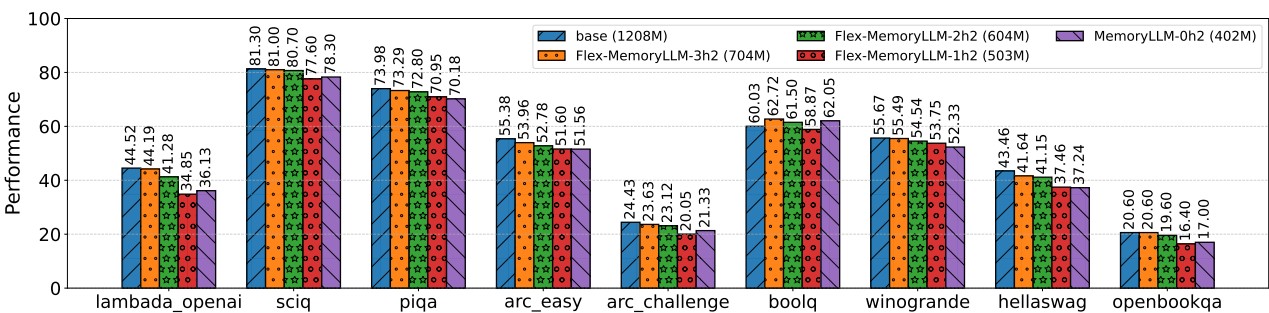

*Figure 8.* Performance comparison of conventional LLM base with MemoryLLM and Flex-MemoryLLM models with ∼ 1B **total** parameter count and varying **active** parameter counts. Training is performed with same recipes & equal token (150B) counts for fairness.

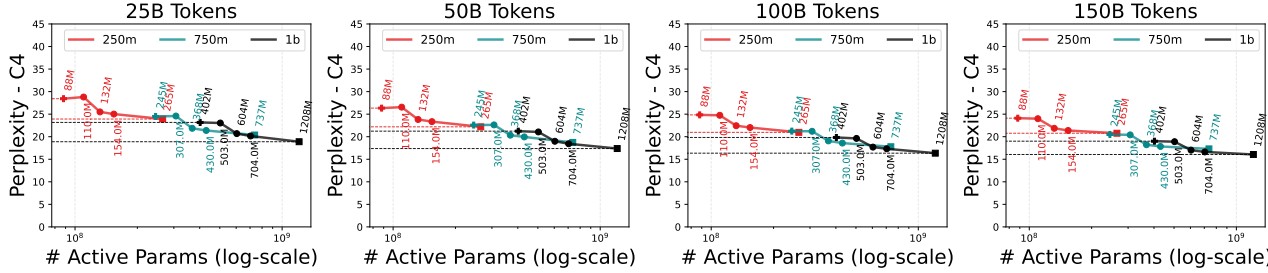

*Figure 9.* Performance comparison of conventional dense LLMs with MemoryLLM and variants of Flex-MemoryLLM at different scale of parameter training tokens and model parameters with same training recipes for fair comparison.

stored in FFN memory from the training data (*e.g.*, wikitext-2, LAMBDA, SiQA, ARC-Easy); (b) tasks that require *logical, causal, or inferential thinking* — where answer isn't directly stored in FFN memory and it must be derived (*e.g.*, HellaSwag, Winogrande, BoolQ, PIQA). To probe the contribution of FFN memory, we decrease $\alpha$ similarly to our prior formulation. Table 1 presents the results of this study of FFNs in a MemoryLLM-1B checkpoint across two task categories for $\alpha \in 1.0, ..., 0.5$, where $1.0$ indicate FFNs contribute as wholly to residual flow. We observe a novel finding: gradually reducing FFNs contribution hurts tasks that heavily rely on recall or retrieval of known information relatively **more than** reasoning or logical thinking tasks.

## 4. MemoryLLM: Performance and Efficiency

### 4.1. MemoryLLM Comparison with Conventional LLM

In section 2.3.2, we discussed how FFNs in MemoryLLM after training can be precomputed and offloaded as ToLs to reduce VRAM requirements as well as computational cost. With two-thirds of *total model parameters* typically occupied by feed-forward modules, this lookup strategy reduces effective *active parameters* in VRAM to be one-third of total model parameters. Although MemoryLLM is designed to improve FFN interpretability, we investigate how MemoryLLM architecture compares to conventional base LLMs on performance standards. To minimize any hyperparameter influence on performance, we used the same training settings for MemoryLLM and base architecture for a fixed

number of training tokens (25-150 billion). For additional implementation details, please refer Appendix A. Table 2 outlines two findings: (1) in reference to total number of parameters, MemoryLLM performance *fall short* of a conventional base LLM counterpart; (2) however in reference to effective active number of parameters (ToLs are not counted as active), MemoryLLM notably outperforms its dense counterpart. These results motivate us to explore methods to bridge the performance gap between MemoryLLM and the base model without compromising the architecture's ability to treat FFNs as context-free, plug-and-play neural memory.

### 4.2. Flex-MemoryLLM: Bridging Conventional LLM and MemoryLLM

In contrast to conventional dense LLM designs where all FFNs parameters are involved in computation over residual flow, MemoryLLM makes a drastic simplification of using entire FFN parameters as static token-level lookup memory over vocabulary. This abrupt simplification limits the computational capability of MemoryLLM leading to lower performance with same training budget in comparison to its dense counterpart of same total parameter count. To address the performance gap, we propose **Flex-MemoryLLM**, which provide a smooth bridge between performance and interpretability. Figure 10 presents the architecture comparison of MemoryLLM & Flex-MemoryLLM with exactly same total number of parameters. As shown in the diagram, Flex-MemoryLLM splits total FFN parameters in MemoryLLM between **two** parts:

- *FFN Compute (FFN-C):* a linear dense module which operates on residual flow and increases the computational capability of MemoryLLM; and

- *FFN Memory (FFN-M):* a context-free neural memory similar to FFNs in MemoryLLM trained with token embeddings with no connection to residual flow.

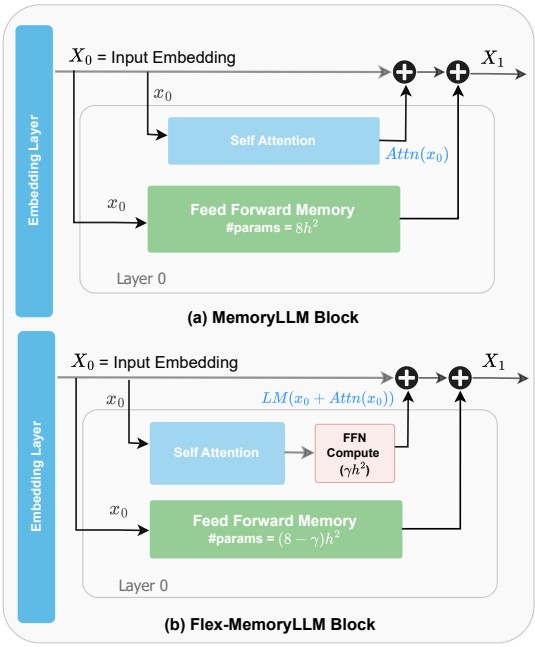

*Figure 10.* Architecture comparison of MemoryLLM & Flex-MemoryLLM with exactly same total number of parameters.

In our experiments, we use the architecture design recipes of LLaMa family models, where feed-forward modules approximately hold $\sim 8h^2$ parameters with intermediate expansion factor of $2.67$, where $h$ represents hidden dimension of the model. To create a Flex-MemoryLLM block, we move $\beta h^2$ parameters from FFN Memory module in MemoryLLM to FFN compute with pursuit to increase the total active parameters of the Flex-MemoryLLM (More details in Appendix A). We experimented with $\beta \in \{1, 2, 3\}$ and we found that $\beta = 3$ can very closely match the performance and training trajectory of conventional base counterpart at the same time enable offloading $5h^2$ parameters as FFN memory from VRAM to storage devices during inference.

Figure 8 presents the performance comparison of conventional base-1B LLM model having 1208M active parameters with respect to three Flex-MemoryLLM-$\beta h^2$ versions with $\beta \in \{1, 2, 3\}$ and MemoryLLM having 704M, 604M, 503M, and 402M active parameter counts respectively. All model checkpoints are trained with exactly same training recipes on 150B tokens for fair evaluation. We observe that, while there exists notable performance

gap between base-1B model and MemoryLLM, the performance gap significantly *diminishes* as we move from MemoryLLM to Flex-MemoryLLM versions. The flexible design Flex-MemoryLLM that split FFN parameters across FFN-C and FFN-M to gradually improve model capacity provides an interesting *balance between efficiency and performance*, having close to base performance with approximately $5h^2$ reduction in active parameters. These results also encourages future studies to closely investigate the over-parameterization ratio of FFNs in modern LLMs and its relationship with training dynamics of LLMs.

In Figure 9, we studied the performance of models with 250M, 750M, and 1B total parameter sizes. The rightmost data point across all three model scale curves represent the conventional base design while leftmost indicate the MemoryLLM. Data points in middle represent three different version of Flex-MemoryLLM-$\beta h^2$ with $\beta \in \{1, 2, 3\}$. We find that: (1) with scaling training from 25B tokens to 150B tokens, the performance gap between MemoryLLM and Flex-MemoryLLM models wrt. base with same total parameter count notably diminishes; (2) ppl difference of Flex-MemoryLLM-$3h^2$ with 704M active parameters closely matches its dense counterpart with 1.2B active parameters; and (3) interestingly, Flex-MemoryLLM-$3h^2$ 1B model with 704M active parameters *outperforms* the base-737M model indicating Flex-MemoryLLM can be a alternative strategy to train superior models with a fixed active parameter count.

### 4.3. MemoryLLM & Flex-MemoryLLM: Alternative Approach wrt. Conventional LLM Pruning

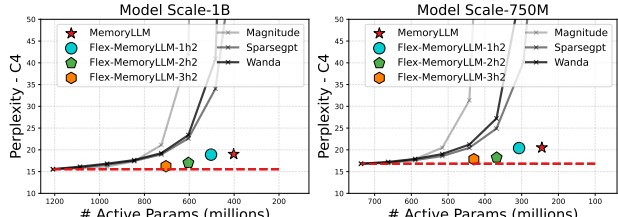

*Figure 11.* Performance comparison of MemoryLLM, Flex-MemoryLLM, and LLM pruning methods wrt. active parameters.

In this section, we investigate how MemoryLLM and Flex-MemoryLLM compare wrt. conventional LLM pruning training techniques, which aim to reduce total active parameter of the model. Figure 11 presents our comparison of MemoryLLM, and Flex-MemoryLLM-$\beta h^2$ built upon 1B and 750M total parameter count wrt. three pruning techniques (Magnitude, SparseGPT, Wanda). The red dotted line indicate the performance of base model with 1B and 750M total and active parameters count. Clearly, we can observe that both MemoryLLM & Flex-MemoryLLM-$\beta h^2$ models are significantly superior compared to training a base model and pruning it to match same active parameter count. These findings illustrate our proposed architecture designs as an

alternative to developing novel pruning techniques.

## 5. Conclusion

We propose MemoryLLM, a modified transformer architecture that explicitly decouples feed-forward networks (FFNs) from the residual stream and self-attention. In our work, FFNs are trained in isolation using context-free token embeddings, enabling their interpretation as neural key-value memory over a finite, human-interpretable query space (the vocabulary). We found that knowledge associated with lexically and semantically similar tokens are indexed across similar memory locations within FFNs. This knowledge is crucial for the performance of retrieval-based tasks. In addition to improved interpretability, this design allows FFNs to be pre-computed as token-wise lookups (ToLs) enabling reduced memory footprint and compute cost.

## Impact Statement

This paper presents work whose goal is to advance the field of Machine Learning. There are many potential societal consequences of our work, none which we feel must be specifically highlighted here.

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

# A. Implementation Details

*Table 3.* Model training configurations for our Base, MemoryLLM, Flex-MemoryLLM Models. All model checkpoints are trained within the paper adopt exactly same configuration for fair comparison.

| Category | Key | Value |
|---|---|---|
| Common | Tokens Count | 25-150 Billion |
| | Vocabulary size | 128,256 |
| | Tokenizer | meta-llama/Llama-3.1-8B |
| | Dataset | C4 |
| | Sequence Length | 2048 |
| | Hidden Activation | SiLU |
| Loss | Name | Cross Entropy |
| | Z-loss | 1.0e-6 |
| Optimizer | Name | Adam |
| | Weight Decay | 0.1 |
| | Beta1 | 0.9 |
| | Beta2 | 0.95 |
| Schedular | Warmup Initial LR | 1e-06 |
| | Warmup Iterations | 5000 |
| | Type | Cosine |
| | Max LR | 1.0e-04 |
| | Min LR | 1.0e-05 |

*Table 4.* FFN parameter division for usage as context-dependent FFN and context-free Memory FFN.

| Architecture | FFN-C | FFN-M |
|---|---|---|
| Base | 33.554M | 0M |
| MemoryLLM | 0M | 33.554M |
| Flex-MemoryLLM-$h^2$ | 4.194M | 29.360M |
| Flex-MemoryLLM-$2h^2$ | 8.388M | 25.165M |
| Flex-MemoryLLM-$3h^2$ | 12.582M | 20.971M |

# B. Understanding Decoding Cost with ToLs

In this section, we aim to investigate how does our proposed architectures MemoryLLM and Flex-MemoryLLM empirically perform wrt. decoding speed (ms/token) which accounts for loading ToLs from storage devices to VRAM. We also enlist the empirically observed inference memory requirements while running our experiments. All experiments are reported using $1 \times$A100 GPU with ToLs stored in BF16 and 2048 sequence length.

# C. Background Work

### C.1. Memory Augmented Architectures

Memory-augmented models are designed to expand a model's effective parameter space without incurring large computational overhead. Early work on memory networks was introduced by (Weston et al., 2014), and later extended with fully end-to-end trainable variants with (Sukhbaatar et al., 2015). Neural Turing Machines (Graves et al., 2014; 2016) incorporate an external, trainable memory that works alongside other neural components to simulate a differentiable, trainable computing system. Product-key networks (Lample et al., 2019) improve the efficiency and scalability of memory retrieval and propose a key-value memory layer that can scale to very large sizes while keeping exact search on the key space. More recently, PEER (He, 2024) has advanced these ideas by replacing traditional vector-based memory values with rank-one matrices, linking memory-augmented architectures with mixture-of-experts models.

Accurate factual generation remains a critical objective for generative models, often evaluated using open-domain question answering benchmarks (Chen et al., 2017; Chen & Yih, 2020) and other tasks requiring substantial knowledge (Petroni et al., 2021). Models that can effectively encode factual knowledge from training data are better equipped to provide correct

*Table 5.* Architecture design of Base, MemoryLLM, Flex-MemoryLLM Models models with different scale in our experiments.

| Total Params | Configuration | Active Params | # Layers | Hidden Dim | Intermediate Dim | #Attn Heads |
|---|---|---|---|---|---|---|
| 250M | Base | 265M | 24 | 960 | 2560 | 16 |
| | Flex-MemoryLLM-$h^2$ | 154M | 24 | 960 | 2240 | 16 |
| | Flex-MemoryLLM-$2h^2$ | 132M | 24 | 960 | 1920 | 16 |
| | Flex-MemoryLLM-$3h^2$ | 110M | 24 | 960 | 1600 | 16 |
| | MemoryLLM | 88M | 24 | 960 | 2560 | 16 |
| 750M | Base | 737M | 24 | 1600 | 4272 | 16 |
| | Flex-MemoryLLM-$h^2$ | 430M | 24 | 1600 | 3738 | 16 |
| | Flex-MemoryLLM-$2h^2$ | 368M | 24 | 1600 | 3200 | 16 |
| | Flex-MemoryLLM-$3h^2$ | 307M | 24 | 1600 | 2668 | 16 |
| | MemoryLLM | 245M | 24 | 1600 | 4272 | 16 |
| 1B | Base | 1208M | 24 | 2048 | 5464 | 32 |
| | Flex-MemoryLLM-$h^2$ | 704M | 24 | 2048 | 3418 | 32 |
| | Flex-MemoryLLM-$2h^2$ | 604M | 24 | 2048 | 4096 | 32 |
| | Flex-MemoryLLM-$3h^2$ | 503M | 24 | 2048 | 4778 | 32 |
| | MemoryLLM | 402M | 24 | 2048 | 5464 | 32 |

*Table 6.* Empirical Memory Requirement and Token Decoding estimated for MemoryLLM and Flex-MemoryLLM variants in comparison to Base transformer model at 1B total parameter scale.

| | Base | Flex-MemoryLLM-$h^2$ | Flex-MemoryLLM-$2h^2$ | Flex-MemoryLLM-$3h^2$ | MemoryLLM |
|---|---|---|---|---|---|
| Inference Memory (GB) | 9.541 | 7.025 | 7.409 | 7.825 | 6.041 |
| Decoding Speed (ms/token) | 21.50 | 18.75 | 20.28 | 21.47 | 14.42 |

responses to knowledge-intensive queries. While larger models generally demonstrate improved factual accuracy (Roberts et al., 2020; Brown et al., 2020), hallucination remains a persistent challenge. One effective approach for mitigating this issue is retrieval-augmented generation, which leverages external knowledge sources to improve factual consistency (Lewis et al., 2020; Karpukhin et al., 2020; Khandelwal et al., 2019). Several language models have incorporated text retrieval from the pretraining stage. REALM (Guu et al., 2020) augments a BERT model with one retrieval step to solve QA tasks. Retro (Borgeaud et al., 2022) enhances auto-regressive decoding with multiple rounds of retrieval, once per 64 tokens. The retrieved texts are injected through a two-layer encoder and then several cross-attention layers in the decoder. Retro++ (Wang et al., 2023a) explores the scalability of Retro by reproducing Retro up to 9.5B parameters. Meanwhile, several models are adapted to retrieval in the finetuning stage. WebGPT (Nakano et al., 2021) learns to use search engine through imitation learning in a text-based web-browsing environment. Toolformer (Schick et al., 2023) performs decoding with multiple tools including search engine, and the finetuning data is labeled by the language model itself.

### C.2. Understanding Feed-Forward Networks in Transformers.

Several studies have investigated the role of feed-forward networks (FFNs) in transformers, particularly their contribution to storing and retrieving knowledge learned during pretraining. (Geva et al., 2021) demonstrated that FFNs can be interpreted as key–value memories that activate on specific lexical or semantic patterns, while follow-up work showed that FFNs promote vocabulary-level concepts during prediction (Geva et al., 2022). Additional related analyses in embedding space further explored how FFN activations correspond to linguistic features and factual recall (Dar et al., 2023; Nichani et al., 2024). Within their framework, the first layer acts as a pattern detector ("keys") while the second layer projects specific information into the residual stream ("values"). This modularity is evidenced by the identification of specific "knowledge neurons" responsible for storing distinct facts. More broadly, the interpretation of neural networks as associative or persistent memory systems connects this line of work to earlier memory-augmented architectures (Sukhbaatar et al., 2019). However, these analyses rely on contextualized residual activations and require extensive post-hoc mining of calibration data, making the inferred query space indirect and difficult to interpret. A recent work, MoLE (Jie et al., 2025), illustrates that in mixture-of-experts (MoEs), majority of experts can be trained directly with token-level input embeddings. However, MoLE's experts computation remains *conditional on contextual information* from routers trained with attention output. In addition, MoLE's success is strongly tied to shared experts trained in conventional fashion with intermediate hidden features as input.

It remains *unclear* if FFN computation within dense LLMs can be disentangled from any intermediate activations without significantly hurting model trainability. In contrast, our work eliminates contextual ambiguity by training dense transformer FFNs directly on context-free token embeddings, enabling deterministic and token-level interpretability.

## D. Understanding Storage Challenges of ToLs

Our proposed MemoryLLM and Flex-MemoryLLM architectures provide an opportunity to pre-compute computationally expensive FFN modules as token-wise lookup tables (ToLs), which can be offload to storage devices in resource-constrained settings. This leads to the question: *How does the trade-off between VRAM and storage devices look like and what can be done to minimize ToLs storage cost?*

We first estimate the total storage cost of LUT as follows:

$$\text{Storage Size} = \text{vocab\_size} \times \text{num\_layers} \times \text{hidden\_dim} \times \text{bits\_per\_param} \tag{11}$$

For our MemoryLLM-1B model with 24 layers and 2048 hidden dimension trained with LLaMa-3.1 tokenizer with 128,256 vocabulary size, $\sim$12.6 GB of storage space is required for ToLs with F16 precision. To address our question, we performed a preliminary investigation[1] of storage challenges of ToLs from **three** different perspectives:

D1. Quantization of token-wise ToLs,

D2. Low Rank compression of token-wise ToLs, and

D3. Layer-wise ToLs compression.

### D.1. Quantization of Token-wise ToLs

*Table 7.* Performance comparison of MemoryLLM-1B with various low-precision token-wise lookup table.

| Precision | Size (GB) | Wikitext-2 ($\downarrow$) | LAMBDA($\uparrow$) | SiQA($\uparrow$) | ARC-Easy($\uparrow$) | HellaSwag($\uparrow$) | Winogrande($\uparrow$) | BoolQ($\uparrow$) | PIQA($\uparrow$) |
|---|---|---|---|---|---|---|---|---|---|
| 16-bit | 12.6 GB | 24.348 | 0.3613 | 0.7830 | 0.5156 | 0.3724 | 0.5233 | 0.6205 | 0.7018 |
| 8-bit | 6.3 GB | 24.347 | 0.3615 | 0.7831 | 0.5156 | 0.3722 | 0.5433 | 0.6206 | 0.7021 |
| 4-bit | 3.15 GB | 24.439 | 0.3610 | 0.7822 | 0.5157 | 0.3700 | 0.5429 | 0.6214 | 0.7011 |

### D.2. Low Rank Compression of Token-wise ToLs

*Table 8.* Performance comparison of MemoryLLM-1B with uniform low-rank SVD compression of ToLs across 24 layers.

| Rank Reduction | Hidden Dim | U #Params | V #Params | Total ToL #Params | Total ToL Size | Storage Reduction % | C4-PPL |
|---|---|---|---|---|---|---|---|
| 0% | 2048 | - | - | 6304.03M | 12.60 GB | 0% | 18.919 |
| 10% | 1843 | 5673.01M | 90.58M | 5763.60M | 11.52 GB | 8.57% | 18.923 |
| 20% | 1638 | 5041.99M | 80.51M | 5122.51M | 10.24 GB | 18.74% | 18.958 |
| 30% | 1433 | 4410.98M | 70.43M | 4481.41M | 8.96 GB | 28.91% | 19.015 |
| 40% | 1228 | 3779.96M | 60.35M | 3840.31M | 7.68 GB | 39.08% | 19.126 |
| 50% | 1024 | 3152.01M | 50.33M | 3202.35M | 6.40 GB | 49.20% | 19.586 |

Several recent works (Li et al., 2023; Wang et al., 2023b; Kaushal et al., 2023) have explored the low-rank characteristics associated with weights and gradients to address storage demands and computational complexity linked to the large matrices of LLMs. For a given transformer layer $L$, the corresponding ToLs with have a dimension of $\text{vocab\_size} \times \text{hidden\_dim}$ represented as $\text{ToL}_L \in R^{|V| \times d}$. A simple SVD decomposition with rank $r$ of $\text{ToL}_L$ will produce two matrices $\text{U} \in R^{|V| \times r}$ and $\text{V} \in R^{r \times d}$ and instead of storing $\text{ToL}_L$, we can store its low-rank representation $(\text{U}, \text{V})$ if $r$ is sufficiently low. We estimate the rank $r$, below which storage of $\text{U}, \text{V}$ will save space as follows:

$$(|V| \times r) + (r \times d) \leq |V| \times d \ \Rightarrow r \leq \frac{|V| \times d}{(|V| + d)} \tag{12}$$

---

[1]An effective novel compression technique for ToLs compression is out of the scope of this work. Our preliminary investigation reveals a high redundancy within the ToLs and leaves sophisticated studies to capitalize these redundancies as future work.

Solving $r$ for MemoryLLM-1B model with 24 layers and 2048 hidden dimension trained with LLaMa-3.1 tokenizer with 128,256 vocabulary size, gives $r \leq 2015$. It implies that if we can save storage space if we can perform $\geq 2\%$ rank reduction within $\text{ToL}_L$. We first start with investigating the low-rank properties within the ToLs of corresponding to different layers. Figure 13 presents the 2048 normalized singular values corresponding to different layers across 24 transformer blocks of MemoryLLM-1B. We observe that the majority of ToLs elicit a heavy tail behavior, indicating better low-rank expressivity. Heavy tails indicate that only a small fraction of singular values carries maximum information and the corresponding matrix can be well approximated using a fraction of basis vectors from SVD with small reconstruction error. In addition, another observation indicate that ToLs of terminal layers tends to have better low-rank properties and friendly to compression in comparison to the middle layers.

For simplicity, we perform uniform SVD on the ToLs across all layers, and report our findings in Table 8. We observe that simple SVD-based low rank compression can significantly reduce ToL storage cost ($\sim 2\times$) with a marginal change in performance of the model. Provided the existence of non-uniform low-rank properties across different layers, we strongly believe that ToLs can be further compressed using non-uniform rank reduction techniques with relatively superior performance compared to uniform SVD.

### D.3. Layer-wise ToLs Compression

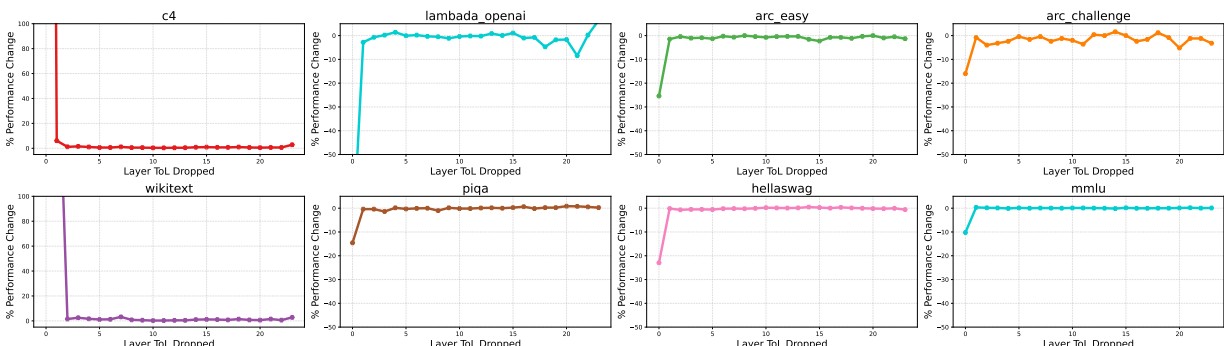

*Figure 12.* Performance change when layer $L$ ToL is dropped in MemoryLLM-1B. Each subplot is a task; within each subplot, the y-axis is the task performance and the x-axis the is the layer of the dropped.

As discussed in Section 2.3.2, the unique design of MemoryLLM allows a plug-n-play design for ToLs usage without any disruption in residual flow. To address the storage challenge of ToLs, we investigate the layer-wise importance of ToLs to validate if ToLs of some layers can be dropped without significant impact on model capabilities. Figure 12 presents performance across 8 tasks when the ToL corresponding to a certain layer $L$ is dropped. Across all the tasks, the major performance degradation comes from dropping ToLs of first few layers. This strongly suggest that majority of ToLs prominently across the middle layers are highly redundant and have marginal impact on performance. Under limited storage availability, dropping middle layer ToLs is a promising compression direction.

## E. Additional Performance Evaluation on MT-Bench

In this task, we investigate *how MemoryLLM and Flex-MemoryLLM perform on open-ended questions and evaluate their multi-turn conversational and instruction-following ability*. To compare the performance, we closely follow the prompt design setting in MT-Bench (Zheng et al., 2023). We rely on the 80 high-quality multi-turn questions that covers common-use human-centric interaction with LLMs, and focus on challenging questions to differentiate models. We used 8 common categories of user prompts to guide the prompt construction to interact with `Base` and our proposed architecture for tasks related to writing, roleplay, extraction, reasoning, math, coding, *etc*.

*Table 9.* Performance comparison of `Base`-1B with MemoryLLM-1B and Flex-MemoryLLM-1B variants on multi-turn conversation across 8 different categories.

| Base | FlexMemoryLLM-3h2 | FlexMemoryLLM-2h2 | FlexMemoryLLM-2h2 | MemoryLLM |
|------|-------------------|-------------------|-------------------|-----------|
| 5.29 | 5.31 | 5.02 | 4.83 | 4.69 |

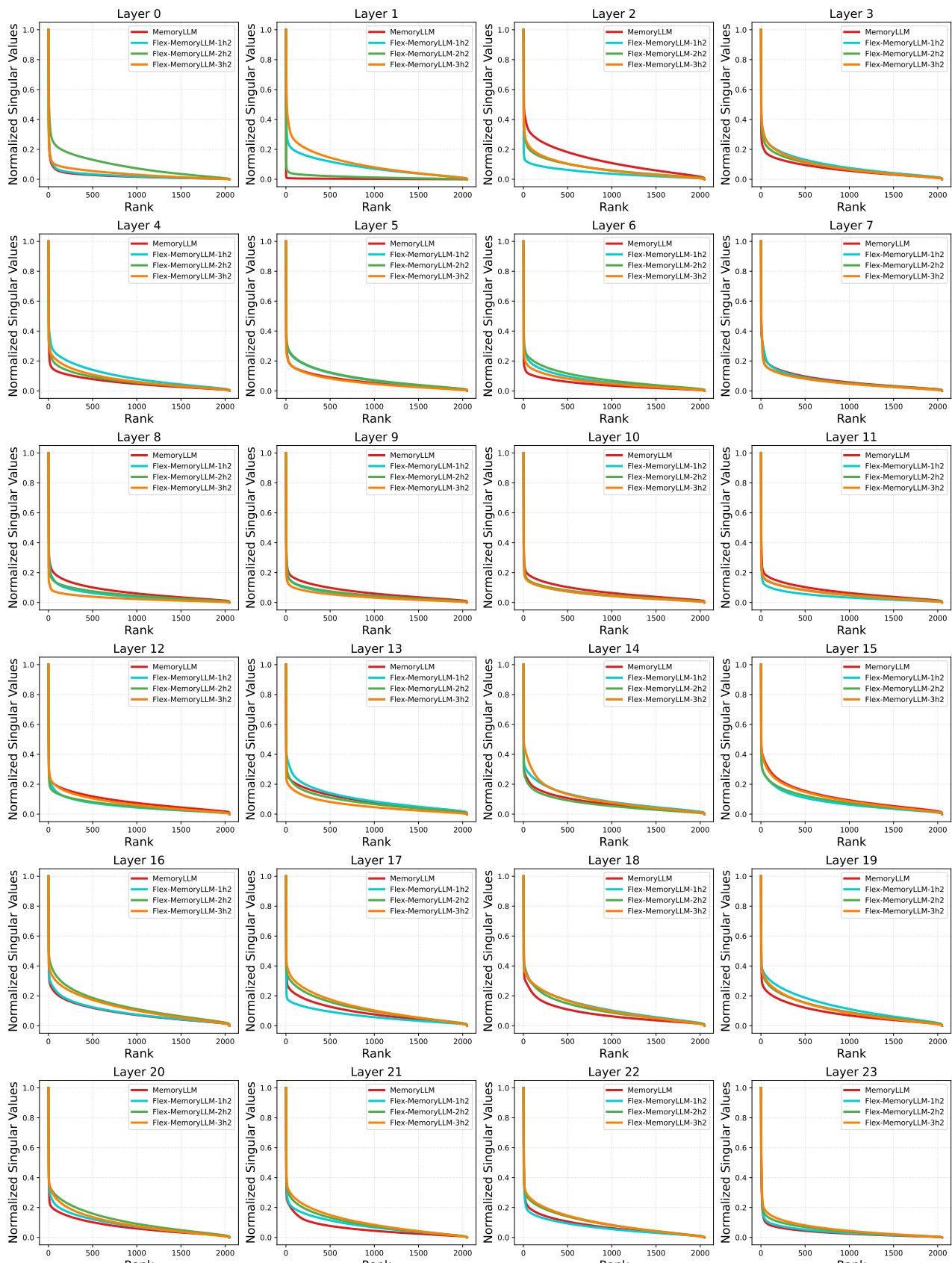

*Figure 13.* Normalized and sorted 2048 singular values of the ToLs corresponding to different 24 layers of MemoryLLM and Flex-MemoryLLM models with 1B scale.

