# OpenReview forum: "MemoryLLM: Plug-n-Play Interpretable Feed-Forward Memory for Transformers"
_ICML.cc/2026/Conference — ICML 2026 regular_

### Official Review · Reviewer_EuUE · 2026-03-09

**Soundness:** 4
**Presentation:** 2
**Significance:** 3
**Originality:** 3
**Overall Recommendation:** 4
**Confidence:** 4

**Summary:**

This paper reexamines the dependency between self-attention and FFNs in LLMs and proposes a decoupled MemoryLLM architecture. Compared to conventional LLM Transformer architectures, MemoryLLM adopts a context-free paradigm where all FFN inputs derive from the initial embedding (along with a LayerNorm) rather than intermediate layer features. This paper also highlights several characteristics of this approach regarding interpretability and efficiency. For instance, while maintaining consistent total parameters, MemoryLLM can operate with fewer active parameters compared to conventional Transformer architectures. Overall, I believe the experiments in this paper are comprehensive, and the design approach is novel and unique.

**Compliance With Llm Reviewing Policy:**

Affirmed.

**Final Justification:**

According to the rebuttal, I will maintain my score.

**Key Questions For Authors:**

For MemoryLLM:

1) Does the active parameter refer to the parameters that need to be stored in VRAM during inference?

2) Are the parameters related to FFN precomputations excluded from the active parameter?

**Limitations:**

Please refer to the Strengths and Weaknesses section.

**Strengths And Weaknesses:**

Soundness:
I believe the experimental design in this paper is quite comprehensive and reliable. After outlining the architectural design methodology of MemoryLLM in Section 2.2, Section 2.3 further elaborates on its interpretability and efficiency. Personally, I am quite impressed with the Plug-n-Play ToL design. Indeed, if all FFNs are placed after the initial embedding, we can first compute and store the corresponding FFN outputs for all vocabulary tokens. The experiments reveal that MemoryLLM's FFN appears more indispensable for retrieval tasks than for inference tasks. I find this observation intriguing, as it highlights how the FFN functions like a memory unit storing knowledge. Beyond this, Flex-MemoryLLM can be viewed as a trade-off between MemoryLLM and conventional LLMs. However, my questions are: 1) If precomputing FFN outputs for all vocabulary words requires substantial storage space, does this introduce additional storage costs? 2) What are the implications (in terms of efficiency and latency) of reducing the number of active parameters in VRAM?

Presentation:
I believe the expression in this article is relatively clear.

Significance:
I believe this research holds significant value. The explainability and efficiency of large language models (LLMs) represent a hot topic with substantial practical applications in production settings. MemoryLLM explores approaches to enhance both explainability and efficiency.

Originally:
I believe this paper offers a novel perspective on FFN research. Beyond architectural modifications, it critically examines the nature of these changes and their impact on performance.

---

> ### Author Rebuttal · Authors · 2026-03-30
>
> We would first like to thank you for your time and regarding our work quite impressive with positive comments such as comprehensive and reliable experiments, high significance and novelty. We would like to address your questions one by one as follows:
>
> ---
>
> > **[Q1] If precomputing FFN outputs for all vocabulary words requires substantial storage space, does this introduce additional storage costs?**
>
> Yes, our approach provides us a unique way to scale LLMs leveraging SSDs. For example, as shown in Table 7, MemoryLLM with additional cost of ~3GB of SSD storage space, can significantly cut down the cost of VRAM (45-55% reduction). For recent SoTA models, where 70-80% parameters are occupied by thick FFN blocks, MemoryLLM and Flex-MemoryLLM design provide a new axis of compression (unlike standard compression techniques like pruning, low-rank, etc.) to run large model with limited resource constraint. For example, LLaMa-13B which can't be run on a consumer grade GPU like RTX4090 in BF-16, can use Flex-MemoryLLM design ($\beta$ = 3) to offload 50-60% of total FFN parameters (refer figure 11) as ToLs on SSD, can perfromance inference on RTX4090 with 2.5KB asynchronous data transfer from SSD (hidden dim for LLama-13B is 5120 x 0.5 byte for ToLs stored as 4-bit).
>
> ---
>
> > **[Q2] What are the implications (in terms of efficiency and latency) of reducing the number of active parameters in VRAM?**
>
> Reduction in the number of active parameters permits running larger models within strict VRAM limitations of consumer grade GPUs. In terms of efficiency, since the FFNs to ToLs computation is one-time and static, it can be pre-computed for all layers for all vocabulary tokens which in turn notably reduces the FLOPs during inference. In terms of latency, access to SSD incurs some data transfer clock time, with appropriate caching mechanism and asynchronous fetching, it can be significantly parallelized.
>
> ---
>
> > **[Q3] Does the active parameter refer to the parameters that need to be stored in VRAM during inference?**
>
> Yes. Active Params count indicate the params counts which will reside in VRAM during inference. In MemoryLLM, after training, the *FFNs can be converted to ToLs* and doesn't require to be in VRAM during inference but can be asynchronously fetch from SSD.
>
> ---
>
> > **[Q4] Are the parameters related to FFN pre-computations excluded from the active parameter?**
>
> Yes. They are excluded.

---

> > ### Author Rebuttal · Reviewer_EuUE · 2026-04-03
> >
> > Thanks for the authors' nice reply. I will maintain my score.

---

### Official Review · Reviewer_82gS · 2026-03-11

**Soundness:** 3
**Presentation:** 3
**Significance:** 3
**Originality:** 3
**Overall Recommendation:** 4
**Confidence:** 4

**Summary:**

The authors propose MemoryLLM, which decouples feed-forward modules from self-attention, treating them as context-free, token-indexed neural retrieval memory to enhance model interpretability. By enabling FFNs to be pre-computed as static lookups, this design significantly improves inference efficiency and offers a flexible performance-efficiency balance through the hybrid Flex-MemoryLLM variant.

**Compliance With Llm Reviewing Policy:**

Affirmed.

**Key Questions For Authors:**

see Weakness

**Limitations:**

see Weakness

**Strengths And Weaknesses:**

**Strength**

1.  By decoupling the information flow of the FFN from the attention module, the investigation into FFN interpretability becomes significantly more convenient and transparent.
2.  The proposed MemoryLLM enables inference using static, context-free lookups, allowing FFNs to be pre-computed and offloaded to storage; this effectively addresses both memory and computational constraints.

3.  The overall presentation and logical flow of the paper are clear, making it highly accessible and reader-friendly.

**Weakness**

1. My primary concern is that while decoupling FFNs from self-attention makes their interpretability more reachable, the fact that FFN inputs are context-free independent tokens may limit MemoryLLM’s language understanding. For instance, in certain scenarios, a single word can have entirely different meanings where the specific interpretation relies heavily on contextual understanding; however, it appears that MemoryLLM offloads this burden to the Attention modules totally, which could potentially restrict the model's overall modeling capacity.

2. As shown in Figure 5, dropping several subsequent layers has a very minimal impact on performance. My concern is whether this implies that repeatedly processing the initial $X_0$​ (which represents relatively shallow, context-free semantic information) in deeper layers is redundant.

3. Regarding the comparison of active parameters, the authors state that "ToLs are not counted as active." I am uncertain whether this is an appropriate distinction. While I understand that ToLs can be pre-computed and do not require real-time computation during the forward pass, from the perspective of the model's representational capacity, there is always a set of underlying "activated" (trained) parameters being utilized—regardless of whether they are viewed as weights within FFN or as dynamic vectors in ToLs. Therefore, the definition of "active parameters" may require further clarification in the main text.

4. I am curious about whether we still need a full dynamic gating mechanism (SwiGLU) in MemoryLLM during training, given that the FFN inputs are relatively static (primarily $X_0$). In conventional LLMs, dynamic gating in FFNs need to handle a vast range of (continuous) inputs, as $X_L$ originates from the attention mechanism's mixture of multiple tokens. However, in MemoryLLM, the inputs of FFNs are finite and discrete (limited to the vocabulary size). Therefore, I wonder if the original dynamic gating in FFNs is still necessary.

   - For example, could we simplify the formulation to: ${\rm FFN}(\hat{X}\_0) = ({\rm seleced} (\gamma) \odot {\rm SiLU}(\hat{X}\_0 W_{\rm up}^T)) W_{\rm down}$  where $\gamma \in R^{\rm Vocab\\_size}$ is a vector controlling the scale and we just need to select the scalars corresponding to $\hat{X_0}$. This could lead to additional savings during training when $d \times 4d >> {\rm Vocab\\_size}$ (for instance, $960\times2560 \approx 2458{\rm k} >> 128k = {\rm Vocab\\_size}$).

   - Alternatively, perhaps even $\gamma$​ is not strictly necessary.

     Of course, such a simplification might also introduce other challenges regarding training optimization or representational capacity. If there are any inaccuracies or oversights, I would be more than happy to discuss them further with the authors.

**typo**

1. In Eq (3) and (5), $W_{\rm down}$ should likely appear on the right side rather than the left，that is， ${\rm FFN}(X_L) = (X_L W_{\rm up}^T \odot {\rm SiLU}(X_L W_{\rm Gate}^T)) W_{\rm down}$ ?
2. In Eq (8), $\tilde{c_{k_i}}$ --> $\tilde{c}_{k_i}$?
3. In Figure 10 (b), $\gamma h^2$ --> $\beta h^2$ ?

---

> ### Author Rebuttal · Authors · 2026-03-30
>
> We would first like to thank you for your time and identifying our work as useful, easy to follow and consisting novel empirical insights. We would now like to address your concerns point-by-point as follows:
>
> ---
>
> > **[W1] Context-free independent tokens may limit MemoryLLM's language understanding**
>
> We fully agree with your concern wrt. MemoryLLM - which is an extreme design choice to feed raw embeddings to FFNs across all layers but to the surprise on positive end (an auxiliary contribution of our work), we showed that such dramatic simplification of transformer architecture can be trained with stability (although with some performance degradation - but not with collapse).
>
> However, specifically to address this gap, we have also proposed Flex-MemoryLLM design - which is position between conventional transformer design and MemoryLLM. Given the prevalent literature around redundancy across FFNs in modern LLMs, Flex-MemoryLLM proposes to divide FFN parameter budget across two parts - (a) FFN-Memory which is fed only the initial context-free embeddings; (b) FFN-Compute which is fed deep attention-mixed residual features like conventional LLMs. It can be clearly observed from our experimental results across Figure 8 and 9, that such hybrid design can closely match the performance of conventional design - while significantly reducing the VRAM requirement of FFNs in conventional design.
>
> An additional note regarding your attention comment (out of the scope of this submission), we indeed have investigated the attention weight matrices of conventional and MemoryLLM transformer block. In our preliminary studies, we have observed two things: (a) MemoryLLM's attention matrices are better converged (lower rank and smaller variance) - a proxy illustrating higher gradient consumption during training; (b) by probing intermediate attention hidden state ($h_{atten}$) with output layer, we found that MemoryLLM attention hidden state is significantly more informative across tokens (generate unique tokens) during autoregressive decoding while in case of base model, it generates repetitive tokens.
>
> ---
>
> > **[W2] Repeated processing the initial context-free semantic information in deeper layers is redundant.**
>
> Thank you for pointing towards it. We, however see this as a positive signal for researchers who would like to explore this further. This provide future works an opportunity to investigate: (a) how can we effectively use this observation to design a mix-up architecture of conventional transformer block and MemoryLLM blocks; (b) this observation is significantly useful in reducing the storage overhead of ToLs - where terminal layers can be stored with higher precision while other can be significantly compressed; (c) unlike current setting, exploration of a dynamic parameter budget for FFNs across different layers during training; (d) designing relevant caching mechanisms to speed up inference; among many others.
>
> ---
>
> > **[W3] Additional clarification for Active Parameters.**
>
> We first start by re-iterating our definitions: a) Total Params is the count to parameters which were involved during training; (b) Active Params count indicate the params counts which will reside in VRAM during inference. In MemoryLLM, after training, the *FFNs can be converted to ToLs* and doesn't require to be in VRAM during inference but can be asynchronously fetch from SSD.
>
> We would like to politely correct the statement in review- "dynamic vectors in ToLs": ToLs are not dynamic but STATIC and calculated once for all the vocabulary for all layers and can be offloaded to SSD. We promise to put explicit effort to make distinction between Active and Total Params very clear in our revised version.
>
> If noted, across all our experiments, we have grouped our results based on total parameter count to ensure fairness and capture the model representational capacity used during training. We also list active params count along with our results to highlight the opportunity of our design to reduce VRAM requirements.

---

> > ### Author Rebuttal · Reviewer_82gS · 2026-04-04
> >
> > Thanks for the authors' response. I will maintain my score.

---

### Official Review · Reviewer_YnRH · 2026-03-12

**Soundness:** 3
**Presentation:** 4
**Significance:** 3
**Originality:** 3
**Overall Recommendation:** 4
**Confidence:** 2

**Summary:**

This paper proposes a modification to transformer architectures MemoryLLM.
For the original Transformer architecture the input of FFN are from self-attention and the residual.
MemoryLLM made a modification, where feed forward part always uses the raw token embeddings as inputs, making each FFN's output a context-free function of tokens. The author claim that FFNs become interpretable and the outputs can be pre-computed to reducing VRAM usage to roughly one-third of total parameters.  Empirical results shows that FFNs matter most for retrieval-heavy tasks and much less for reasoning tasks, which says that FFN can perform knowledge storage. Furthermore, a hybrid variant flex MemoryLLM is also proposed, which have close performance to original Transformer architecture while still have the efficiency and interpretability benefits.

**Compliance With Llm Reviewing Policy:**

Affirmed.

**Key Questions For Authors:**

The TKV framework evaluates FFNs by feeding token embeddings as queries, and MemoryLLM is specifically designed and trained to receive exactly those token embeddings. THis makes interpretable results guaranteed by construction rather than discovered. Can the authors provide evidence that the memory patterns revealed by TKV are consistent or transferable to standard pretrained models like LLAMA or GPT, where FFNs were not constrained this way? If such correspondence can be demonstrated, it can strengthen the claim that MemoryLLM is a general diagnostic tool for understanding FFNs broadly, rather than self-validating where the architecture and evaluation framework are designed to confirm each other.

**Limitations:**

Yes

**Strengths And Weaknesses:**

Strength:
The logical flow is clear and easy to follow, while there are also figures to illustrate the architectures clearly.
The core idea is simple yet effective. A simple architectural change enables both interpretability and efficiency simultaneously.
The author proposed TKV framework for analysing the interpretability, which improves existing methods.
The finding that FFNs matter more for retrieval tasks than reasoning tasks is a useful and novel empirical insight that helps understand Transformers better.


Weakness:
Since MemoryLLM is a modification of the original Transformer arhcitecture. All the properties and results are based on this modified architecture. It seems hard to carry over the conclusions to the original Transformer architectures. A main gap would be deeper layers, for deeper layers. the inputs of the FFN would be very different from MemoryLLM, which is only the raw embedding tokens.

The performance gap between MemoryLLM and the original baseline at the same total parameter count is a real cost that the paper seems only comparing on active parameters instead

The paper trains only up to 1B parameters on 150B tokens, which is somehow small, it is meaningful to see on larget parameter settings.

---

> ### Author Rebuttal · Authors · 2026-03-30
>
> We would first like to thank you for your time and identifying our work as useful, easy to follow and consisting novel empirical insights. We would now like to address your concerns point-by-point as follows:
>
> ---
>
> > **[W1] MemoryLLM with Raw Embeddings and Deeper Layers**
>
> In our submission, we have explored 3 parameter settings 250M, 750M, and 1B with 24 layers (reasonable depth wrt. parameter count) and found consistent interpretability and performance trends.
>
> We fully agree with your concern wrt. MemoryLLM - which is an extreme design choice to feed raw embeddings to FFNs across all layers but to the surprise on positive end (an auxiliary contribution of our work), we showed that such dramatic simplification of transformer architecture can be trained with stability. However, specifically to address this gap, we have also proposed Flex-MemoryLLM design - which is position between conventional transformer design and MemoryLLM. Given the prevalent literature around redundancy across FFNs in modern LLMs, Flex-MemoryLLM proposes to divide FFN parameter budget across two parts - (a) FFN-Memory which is fed only the raw embeddings; (b) FFN-Compute which is fed deep attention-mixed residual features like conventional LLMs. It can be clearly observed from our experimental results across Figure 8 and 9, that such hybrid design can closely match the performance of conventional design - while significantly reducing the VRAM requirement of FFNs in conventional design.
>
> ---
>
> > **[W2] Paper Comparison is based only on Active Parameter Count**
>
> Thank you for raising this question. First, we would like to clarify the difference between total params and active params (if there exist any confusion) - (a) Total Params is the count to parameters which were involved during training; (b) Active Params count indicate the params counts which will reside in VRAM during inference. In MemoryLLM, after training, the *FFNs can be converted to ToLs* and doesn't require to be in VRAM during inference but can be asynchronously fetch from SSD.
>
> Now back to your concern, we would like to clarify that majority of our experiments in our submission are based on Total number of parameters to provide a fair picture: (a) Figure 7 enlist the performance of Base and MemoryLLM with same 1B param count; (b) Table 2 presents both Active and Total Params count for the readers for clarity; (c) Figure 8 presents performance comparison of Base, MemoryLLM and Flex-MemoryLLM versions - all with exactly same total 1B parameter count; (d) In Figure 9, Red, Aqua, and Black illustrate models with 250m, 750m, and 1B total parameter count.
>
> Note that across, all our experimental results, we intentionally provide active params count to highlight the efficiency benefits of our proposed architecture designs.
>
> ---
>
> > **[W3] Small Training with 1B parameters and 150B Tokens**
>
> We appreciate your concern. During the rebuttal phase, we have been able to extend our 1B model checkpoint training from 150B tokens to $\sim$ 250B tokens. From the interpretability side, the average CC across layers increased from 0.9332 $\rightarrow$ 0.9389 which indicate improved clustering properties of the memory keys. From the performance side, the C4 PPL gap between the performance of conventional transformer wrt. MemoryLLM and Flex-MemoryLLM-3h2 improved by 0.2166 and 0.3711. To further address this feedback, we are also working towards launching large scale experiments for a 3B model with 500 billion token count and we plan to include additional results in the final camera ready submission.
>
> ---
>
> > **[Q] Transferability of TKV Framework**
>
> This is a great question. Note that the architecture of FFNs is unchanged across conventional and MemoryLLM and thereby TKV can be applied to conventional model checkpoints. To understand further, using a validation set of 10 sentences from wikitext-2, we captured the hidden state from the residual stream for individual tokens, and enlist $c_k$ vectors similar to MemoryLLM. For top-50 tokens we found CC = 0.76 while MemoryLLM have a CC = 0.98 corresponding to FFNs with Transformer Layer 1. It clearly illustrate that even conventional transformer design show memory patterns distributed across FFN channels, but due to contextual mixing/flattening - the patterns are not as well elicited as in MemoryLLM.

---

> > ### Author Rebuttal · Reviewer_YnRH · 2026-04-03
> >
> > The author has resolved my major concern regarding the transferability of TKV framework, I will keep my score.

---

### Official Review · Reviewer_XH7f · 2026-03-13

**Soundness:** 3
**Presentation:** 4
**Significance:** 3
**Originality:** 3
**Overall Recommendation:** 5
**Confidence:** 4

**Summary:**

The authors propose a new way to understand and restructure the feed-forward network (FFN) component in transformer-based LLMs. The key idea is to decouple FFNs from the self-attention mechanism, treating them as a form of context-free token-wise neural memory rather than purely computation layers. The FFN branch acts as a residual connection at each transformer layer and is based on using token embeddings so that they act as retrieval memory storing associations between tokens and internal representations.

This design allows FFN activations to be precomputed as token-wise lookups, enabling more interpretable analysis of how tokens access memory locations and potentially improving inference efficiency by moving these lookups between VRAM and storage as needed. The paper also introduces Flex-MemoryLLM, an intermediate architecture that reintroduces limited interaction between attention and FFNs to mitigate performance losses caused by fully decoupling them. Through experiments across downstream tasks, the authors analyze how FFN memory is accessed and evaluate the trade-offs between interpretability, efficiency, and model performance.

**Compliance With Llm Reviewing Policy:**

Affirmed.

**Final Justification:**

I think the quality of the manuscript with the additional experimental results presented in the rebuttal push it to the acceptance bar for me. I have increased my score to an accept.

**Key Questions For Authors:**

- For their FlexMemory setup, can the authors comment on whether the choice of the tradeoff $\beta$ is task-dependent. How would a practitioner go about this choice.

- How stable are the learned memory locations across training data subsets from the same base dataset? I understand the training infrastructure for models at the scale of the experiments in the paper are daunting, but perhaps the authors have conducted smaller scale validation to ascertain this.

- Can the memory interpretation of FFNs help identify or edit specific knowledge stored in the model? On the flip side, do the authors have insights on the category of tasks that may not benefit from the proposed formulation.

**Limitations:**

- Performance trade-offs: Fully decoupling FFNs from attention may degrade performance, and the paper then relies on the workaround based on Flex-MemoryLLM to recover accuracy. It is unclear how to reconcile this within the current ecosystem of training large foundation models (particularly outside of the applications this paper explores)

**Strengths And Weaknesses:**

STRENGTHS:

- The proposed approach treats FFNs as plug-and-play components, allowing them to be trained or modified independently of the attention layers, which is an interesting premise that can potentially offer new insights for several applications

- Precomputing FFNs as token-wise lookups may reduce inference cost and enable flexible memory management, which is a very important direction of research as models scale and training costs explode.

- The experimental analysis is pretty detailed and really well presented. The FlexMemoryLLM architecture seems to offer a viable  compromise between the interpretability of MemoryLLM and performance, while taking steps to address limitations of fully decoupled FFNs.


WEAKNESSES:

- Although interpretability is scattered throughout their motivation, the authors do not venture too much into this space beyond t-sne visualisations.

- Experiments are performed on two categories of tasks, retrieval based and reasoning based tasks. It is unclear whether the trends they observe in terms of the flex-memory tradeoff, pure MemoryLLM against the base models will still hold for other tasks, especially tasks requiring creativity rather than pure lookup

---

> ### Author Rebuttal · Authors · 2026-03-30
>
> We would first like to thank you for your time and identifying our work as important research direction, interesting, and well presented. We would now like to address your concerns point-by-point as follows:
>
> > **[W1] Limited interpretability beyond t-SNE**
>
> We respectfully like to clarify that interpretability exploration of MemoryLLM extend beyond t-SNE:
>
> - We explored the clustering coefficient (CC) of FFN memory key vectors across layers (Figure 6b) to provide a quantitative estimate of how well semantically similar tokens attempt to access similar memory regions within FFNs. Existence of high CC across all layers illustrate that spatial distribution of neural memory in FFNs is consistently built across all the layers.
>
> - We explored the functional importance of token-indexed FFN memory blocks, and empirically validate that it plays a dominant role for tasks which heavily rely on recall or retrieval of known knowledge more than task which require logical, casual, or inferential thinking.
>
> - Furthermore, we have done some qualitative token-level studies during rebuttal in response to your feedback, and discovered that due to additive nature of token-level context-free knowledge in residual flow, MemoryLLM is significantly able to address the issue of smoothening/flattening of intermediate hidden states. For example:
>   - Sentence1 : John and Matt have worked together for 20 years in same company and share good relationship.
>   - Sentence2 : John and Matt have worked together for 20 years in same company and share bad relationship.
>   - $cosine_{memoryllm}$ (s1, s2) = 0.7629 &nbsp;&nbsp; v/s &nbsp;&nbsp; $cosine_{base}$ (s1, s2) = 0.9345
>   - Lower cosine similarity indicate superior ability to capture the difference between good v/s bad.
>
> - In addition, some parallel recent work from Deepseek (https://arxiv.org/abs/2601.07372) and Meta (https://arxiv.org/abs/2601.10639) illustrate that such design choice is highly effective for knowledge editing (another angle for interpretability) and we are currently exploring in that direction with potential to include it in our camera-ready version.
>
> ---
>
> > **[W2] Additional experiments for creativity tasks**
>
> We have conducted addition experiments on MT-Bench (https://arxiv.org/abs/2306.05685) which is multi-turn benchmark (writing, roleplay, extraction, reasoning, math) and it can be clearly observed that our results aligns well wrt. results reported in submitted version.
>
> | Base | FlexMemoryLLM-3h2 | FlexMemoryLLM-2h2 | FlexMemoryLLM-2h2 | MemoryLLM |
> |------|-------------------|-------------------|-------------------|-----------|
> | 5.29 | 5.31              | 5.02              | 4.83              | 4.69      |
>
> ---
>
> > **[Q1] Author's comment on whether the choice of the tradeoff is task-dependent**
>
> For our Flex-Memory setup, while tradeoff can be task-dependent where factual task requiring usage of all FFN-Memory and reasoning-oriented tasks using fewer FFN-Memory; we primarily consider the tradeoff to be resource-dependent. For recent SoTA models, where 70-80% parameters are occupied by thick FFN blocks, Flex-MemoryLLM design provide a new axis of compression (unlike standard compression techniques like pruning, low-rank, etc.) to run large model with limited resource constraint. For example, LLaMa-13B which can't be run on a consumer grade GPU like RTX4090 in BF-16, can use Flex-MemoryLLM design ($\beta$ = 3) to offload 50-60% of total FFN parameters (refer figure 11) as ToLs on SSD, can perform inference on RTX4090 with mere 2.5KB asynchronous data transfer from SSD (hidden dim for LLama-13B is 5120 x 0.5 byte for ToLs stored as 4-bit).
>
> ---
>
> > **[Q2] Stability of Learned Memory Location with data subsets**
>
> Nice question. To address this, we subsampled 50B tokens from our base datasets with different two different seeds to trained our 1B model and found no impact on model performance (PPL 27.258 vs PPL 27.224). In addition, we looked at the cosine similarity of the FFN memory key ($c_k$) vectors across all 24 layers for our 128K vocabulary tokens. Across all layers, we found the mean cosine similarity $\sim 0.91$; and more specifically between Layer 0 - 8 (range in 0.93-0.98), Layer 8-16 (range 0.85 - 0.91), Layer 16-24 (range 0.94 - 0.99) which support the stability of learned memory location.
>
> > **[Q3] Memory Interpretation and Knowledge Editing.**
>
> This is a great question and our next iteration of this project involves exploring how can we utilize our proposed architecture for a more controlled knowledge editing. Some recent parallel works from Deepseek (https://arxiv.org/abs/2601.07372) and Meta (https://arxiv.org/abs/2601.10639) have explored this, they found that it work very well.

---

> > ### Author Rebuttal · Reviewer_XH7f · 2026-04-03
> >
> > Thank you for the detailed responses. My concerns have been sufficiently addressed. Good luck with the submission.

---

### Decision · Program_Chairs · 2026-04-30

**Decision:**

Accept (regular)

**Comment:**

This paper proposes MemoryLLM, an architectural redesign that decouples FFNs from self‑attention and interprets them as context‑free, token‑wise neural memory. By precomputing FFN outputs as token‑wise lookups, the approach offers improved interpretability and significant inference efficiency gains, while the Flex‑MemoryLLM variant recovers most of the performance of standard Transformers.

Reviewers are in general supportive, highlighting the clear motivation, novel design, and strong empirical analysis across retrieval, reasoning, and multi‑turn tasks. Concerns about performance degradation from full decoupling, task generality, and transferability are well addressed through Flex‑MemoryLLM, additional experiments, and thorough rebuttals. The work provides a compelling empirical insight that FFNs function more as knowledge storage than reasoning engines.

Overall, this is a solid and practically relevant contribution. I recommend Accept.